# Nitrate, Auxin and Cytokinin—A Trio to Tango

**DOI:** 10.3390/cells12121613

**Published:** 2023-06-13

**Authors:** Rashed Abualia, Stefan Riegler, Eva Benkova

**Affiliations:** 1School of Plant Sciences and Food Security, Tel Aviv University, Ramat Aviv, Tel Aviv 69978, Israel; 2Institute of Science and Technology Austria, Am Campus 1, 3400 Klosterneuburg, Austria

**Keywords:** nitrate, auxin, cytokinin, crosstalk

## Abstract

Nitrogen is an important macronutrient required for plant growth and development, thus directly impacting agricultural productivity. In recent years, numerous studies have shown that nitrogen-driven growth depends on pathways that control nitrate/nitrogen homeostasis and hormonal networks that act both locally and systemically to coordinate growth and development of plant organs. In this review, we will focus on recent advances in understanding the role of the plant hormones auxin and cytokinin and their crosstalk in nitrate-regulated growth and discuss the significance of novel findings and possible missing links.

## 1. Introduction

As a major component of vital macromolecules such as nucleic acids, amino acids, and chlorophyll, nitrogen is an essential macronutrient for plants [1]. Although nitrogen is one of the most abundant elements in nature, accounting for about 70% of atmospheric gasses, its availability for plant uptake in the soil varies temporally and spatially [2]. Therefore, modern agriculture relies heavily on nitrogen fertilization to maximize crop quality and yield. However, much of this nitrogen leaches into the environment and pollutes water resources with serious environmental and economic consequences [3,4]. Consequently, understanding nitrogen-regulated plant growth and ultimately improving nitrogen use efficiency are the focus of studies aimed at sustainable and efficient agricultural practices [2,4].

Plants take up nitrogen from the soil in inorganic forms, such as nitrate and ammonium, or in organic forms, such as amino acids and peptides. Nitrate is the predominant form of nitrogen in aerobic soils [5] and the preferred nitrogen source for most higher plants, including *Arabidopsis thaliana* [6,7]. The acquisition of scarce nutrients such as nitrogen from the soil is one of the most challenging aspects of plant adaptation to a sessile lifestyle. Plants must cope with the varying availability and source compounds of this element and ensure its optimal uptake into the plant body. This is achieved by adjusting mechanisms and pathways that mediate soil exploration, nitrogen uptake, and distribution within the plant body [8]. The vital function of effective soil utilization and balanced uptake of the nitrogen-containing compounds is executed by the root organ. In the soil, the root system perceives and integrates local and systemic signals about the nitrogen status of the plant to regulate the uptake and distribution of nitrogen. An important component of this nutrient management strategy is flexible modulation of the root system architecture.

For example, after a period of deficiency, nitrate supply stimulates primary and lateral root growth and expansion [9,10,11,12], while supra-optimal nitrate levels have a negative effect on primary and lateral root growth [13,14,15]. The lateral roots of plants growing under heterogeneous nitrate conditions preferentially expand and colonize nitrate-rich zones [13]. Ammonium as the sole source of nitrogen suppresses the growth of primary and lateral roots [15] whereas L-glutamate as an organic nitrogen source inhibits the growth of primary roots but stimulates the growth of lateral roots [16]. This exceptional plasticity of the root system is at the core of nitrogen foraging, the ability of the root to adjust its growth and development to maximize nitrogen uptake under low and fluctuating nitrogen conditions.

Numerous recent studies demonstrated that adaptation responses driven by nitrate/nitrogen are fine-tuned in concert with phytohormones, the endogenous signaling molecules that coordinate nearly every aspect of plant growth and development. Hormone metabolite profiling [14,17,18,19,20], as well as a spectrum of omics approaches [21,22,23,24,25] clearly indicated close interactions between hormonal regulatory networks and pathways controlling nitrogen status. The expression of genes involved in biosynthesis, metabolism, transport, or signal transduction of plant hormones such as auxin, cytokinin, ethylene, abscisic acid, and gibberellins are rapidly modulated in plants exposed to fluctuating nitrogen conditions. NIN-like protein 7 (NLP7), the nitrate master regulator [22,24,26,27] and a recently reported intracellular nitrate sensor [28], was found to regulate components of hormonal regulatory networks [22,24,27,29]. NRT1.1, a well-established nitrate transceptor, was shown to adjust levels and distribution of auxin to low nitrate levels by regulating its biosynthesis and transport [30,31,32]. The inhibitory effects of excessive nitrate supply on root growth and branching were associated with an increase in abscisic acid and ethylene biosynthesis via increased expression of the corresponding biosynthetic genes such as *ABA1*, *ABA2*, and *ABA3* [33] and *ACS* [14]. These examples demonstrate that plant hormones are important endogenous integrators and translators of nitrogen status to plant adaptive responses.

Over the last years, significant progress has been made towards dissecting the functions of hormones in plant adaptations to nitrogen conditions at the molecular level. This review focuses primarily on the interactions between nitrate-related pathways and the regulatory networks determining the activities of auxin and cytokinin, the two plant hormones with key functions in fundamental biological processes such as cell division and differentiation.

## 2. Nitrate Transport and Signaling

To cope with the fluctuating availability and distribution of nitrate in the soil, plants have evolved sophisticated mechanisms to balance nitrate uptake with growth requirements [34,35]. To efficiently acquire nitrate from the soil and distribute it between uptake/storage and sink organs to regulate their nitrogen status. Plants use different transporters with different transport directions, affinities, and specificities to accomplish these tasks [36]. In *Arabidopsis*, proteins of four families have been shown to function as nitrate transporters: NRT1/PTR (nitrate transporter 1/peptide transporter family, 53 members), NRT2 (seven members), CLC (chloride channels, seven members), and SLAC1/SLAH (slow anion channel-associated 1 homologues, five members). Together, these four families comprise 72 genes, although transport activity is not yet reported for all members [37,38]. Nitrate transporters are divided into influx and efflux transporters [39], with the former being involved in the uptake of nitrate from the soil. While the role of efflux transporters is not yet fully understood, their function in the uptake of nitrate into the xylem/phloem has long been suspected [40]. Since nitrate is an important but limited resource, plants require mechanisms for efficient uptake of nitrate. To cope with highly variable nitrate conditions in the soil, plants rely on transporters with different properties to maximize their nitrate uptake capacity. Nitrate influx transporters belong to two different systems classified according to their efficiency in nitrate transport, the Low Affinity Transport System (LATS) and the High Affinity Transport System (HATS) [6]. The LATS facilitates transport at high (greater than 0.5 mM) external nitrate concentrations, whereas the HATS mediates uptake at low (less than 0.5 mM) external nitrate concentrations [37,41,42,43,44]. The transport activity of these systems depends on the cellular energy supply and is coupled to the electrochemical gradient of protons [45]. An additional level of complexity is added by the regulation of gene expression of high-affinity transporters. Nitrate supply was found to regulate the expression levels of several high-affinity transporters, therefore termed induced high-affinity transporters (iHATS), while the others are referred to as constitutively expressed transporters (cHATS) [6,42,44].

A major nitrate transporter from the NRT1/PTR family reflects the importance of nitrate to plants and the complexity of its transport system. Under low-nitrate conditions, NRT1.1 upon CIPK-CBL9 (CBL-Interacting Protein Kinase, CBL:Calcineurin-B like protein) mediated phosphorylation switches from a low-affinity to a high-affinity nitrate transporter [46,47]. Notably, NRT1.1, is not only a dual affinity nitrate transporter [46,48,49,50,51], but acts also as a nitrate sensor [52]. After NRT1.1 senses the provision of nitrate, it activates calcium-dependent protein kinases (CPKs) which in turn phosphorylate the transcription factor NIN-like protein 7 (NLP7). Following, phosphorylation NLP7 moves to the nucleus, where it enhances expression of nitrate-regulated genes, including components of nitrate transport and assimilation networks [24,28,29]. NLP7 requires activation by CPKs along with its own ability to recognize nitrate to activate primary nitrate responses that affect organ growth and architecture [24,28,29]. Moreover, the transport function of NRT1.1 is also involved in the regulation of genes such as *AFB3* and *NAC4*, independently of calcium-mediated signaling and NLP7 [11,12,24,29]. This suggests the existence of other intracellular mechanisms for nitrate signal transduction that are still unknown.

Another important subgroup of nitrate transporters is the *NRT2* gene family. However, to transport nitrate, those transporters require interaction with Nitrate Assimilation Related Protein 2 (NAR2) [53]. NRT2.1 is the primary transporter of this family responsible for about 75% of nitrate influx into the root. It is a high-affinity nitrate transporter localized to the plasma membrane of epidermal and cortical cells of the roots [41,44,54]. Similar to NRT1.1, NRT2.1 has also been proposed as a nitrate signal transducer because of its role in lateral root development on nitrate, but its specific molecular signaling pathway remains to be elucidated [55,56].

While these early responses are critical for adaptive responses to nitrate, later responses depend on subsequent gene expression and protein synthesis to regulate feedback to nitrate uptake, assimilation, and adaptive responses through local and systemic signaling. Transcriptomic approaches allowed for the identification of several transcription factors involved in nitrate responses, such as TGA1 and TGA4. The *tga1tga4* double mutant showed reduced lateral root initiation and elongation in response to nitrate, suggesting that these transcription factors play an important role in the response of root development to nitrate [10]. For a detailed overview of nitrate transport, sensing, and signaling, we refer the reader to comprehensive reviews on these topics [35,37,57].

## 3. The Role of Auxin in Nitrate-Regulated Plant Growth and Development

Auxins are a group of naturally occurring molecules derived from tryptophan, with indole-3-acetic acid (IAA) being the major form of auxin. The biosynthesis of IAA is defined by a two-step metabolic pathway, in which the TAA family of aminotransferases converts tryptophan (Trp) to indole-3-pyruvate (IPA), followed by a YUC flavin monooxygenases-mediated conversion of IPA to IAA [58].

Auxin has extensive regulatory functions in plant development, including tropic responses, embryogenesis, and postembryonic initiation and formation of organs [59,60,61]. The auxin signal transduction cascade is activated by the hormone-triggered interaction of the auxin receptor SCFTIR1/AFB E3 ubiquitin ligase with Aux/IAA signaling repressors, which leads to the latter’s polyubiquitination and degradation by the proteasome. Consequently, transcription factors of the Auxin Response Factor (ARF) family are relieved from inhibition by Aux/IAAs and transcription of auxin-responsive genes is promoted [62].

In *Arabidopsis*, there are 23 ARFs displaying differential affinities to members of the Aux/IAA repressor family, which encompasses 29 homologues [62,63]. Variable homo- and hetero-oligomerizations of Aux/IAAs may provide an additional mechanism for the diversity of the auxin response [63,64].

Beyond the canonical auxin signal transduction cascade, revolving around TIR1/AFB-Aux/IAA-ARF, observations of auxin-triggered rapid non-transcriptional growth responses suggest another auxin receptor/sensor might operate *in planta* [65,66]. Recently, ABP1 and the auxin signaling proteins of the transmembrane kinase (TMK) family were shown to interact with plasma membrane H+-ATPases, inducing their phosphorylation and thereby promoting cell wall acidification and rapid elongation of hypocotyl cells in *Arabidopsis* [67,68,69].

Besides auxin metabolism, perception and signal transduction, the tightly controlled transport machinery is another key component of the regulatory system determining the biological activity of auxin. In higher plants, auxin is transported from young leaves to roots via the phloem vasculature [70]. This long-distance auxin transport is complemented by polar auxin transport (PAT), mediating cell-to-cell transport of the hormone [71,72]. This slower mode of auxin transport depends on active auxin influx and efflux between cells and is of great biological importance as it enables the directional movement of auxin as well as distribution gradients across tissues and organs. PAT is mediated by several families of membrane transporters including AUX1/LIKE AUX (AUX/LAX), PIN-FORMED(PIN), PIN-LIKES (PILS), and ATB Binding Cassette B (ABCB) [30,73,74,75,76].

Considering the importance of auxin in plant growth, developmental and physiological processes, it is not surprising that the investigation of its role in adaptation to nitrogen sources and in particular to nitrate availability has become one of the major research foci over the last decades. Early experiments conducted in the 1930s and 1940s showed that the auxin content in shoots of *Brassica caulorapa* and other species is dependent on the amount of supplied nitrate [77]. Since then, numerous works have pointed out that auxin biosynthesis, transport and signaling pathways are important mechanisms underlying plant growth and developmental adaptation to varying levels and sources of nitrogen [9,19,30,31,32,78,79,80]. A study by Ma et al. [81] showed that expression of key components of auxin biosynthesis including *tryptophan aminotransferase* 1 (*TAR1*), *TAR2*, and their close homologs *TAA1* is regulated by nitrogen availability. Among them, TAR2 was found to play a critical function in maintaining auxin levels and fine-tuning lateral root outgrowth under mild nitrogen-limiting conditions [81]. *TAR2* expression is controlled by NRT1.1, which acts as a negative regulator under nitrate depletion conditions. The suppression of *TAR2* transcription is abolished either by the provision of nitrate or in *nrt1.1* mutant [31]. Collectively, these studies demonstrate how nitrate contributes to fine-tuning lateral root outgrowth and adjusting it to fluctuating nitrate availability via TAR2-mediated biosynthesis of auxin in the root stele.

Identification of several components of the PAT machinery including *PIN1*, *PIN2*, *PIN4* and *PIN7* in the nitrate-responsive transcriptome suggested that the distribution of auxin in the plant body is controlled by nitrate [25]. This conclusion has been confirmed by Maghiaoui et al. [31], who demonstrated that mRNA levels of *PIN1*, *PIN4*, *PIN7*, but also *ABCB4*, *ABCB19* auxin transporters are modulated by nitrate—independently of NRT1.1 perception however, thus raising a question about the molecular bases of this regulatory network. Nitrate-regulated transcription of auxin influx carriers such as *AUX1* and *LAX3* on the other hand, is dependent on NRT1.1 [31] and plays an important role in adjusting lateral root outgrowth to nitrate availability. Intriguingly, in addition to the well-established components of PAT such as PINs, AUX/LAX and ABCB transporters also NRT1.1, initially identified as a dual nitrate transporter, was found to transport auxin [30,82]. The auxin transport activity of NRT1.1 turned out to be particularly important for adjusting root branching to nitrate availability. Under low nitrate conditions, NRT1.1 transports auxin away from the tip of the lateral root primordium (LRP), which ultimately results in its developmental arrest [30]. Taken together, the NRT1.1 transceptor coordinates auxin-dependent development of LRPs via local control of auxin synthesis, redistribution of auxin in the primordium, and fine-tuning expression of *LAX3* in the tissue overlying the LRP. There, the LAX3 influx driven accumulation of auxin controls cell wall loosening which allows the LRP to emerge through adjacent tissues (Figure 1A) [30,31,83,84].

Global scale proteome and phosphoproteome analyses in *Arabidopsis* revealed that nitrate provision to nitrogen starved plants triggers rapid changes in protein phosphorylation [23]. Among those proteins whose phosphorylation status is altered in response to nitrate provision, the PIN2 auxin transporter was recovered. Nitrate-specific PIN2 phosphorylation sites were shown to determine the membrane localization of this auxin transporter in epidermal and cortex cells at the root apical meristem. The fine-tuning of PIN2 levels at the plasma membrane by nitrate coordinates the auxin distribution between two adjacent cell files and thereby primary root growth. Hence, posttranslational regulation of auxin transport by nitrate enables the altering of auxin fluxes to rapidly modulate root growth. However, the kinases and phosphatases involved in this process are yet to be identified (Figure 1B) [9,23].

The canonical nitrate signaling cascade involves NRT1.1-mediated activation of calcium-dependent signaling via calcium dependent protein kinases CPK10, CPK30, and CPK32, which phosphorylate and activate NLP7, the nitrate sensor and master regulator of the nitrate response [24,26,27,28,29]. Notably, another member of the CPK family, CPK29, was found to interact with PIN1 and PIN3 auxin transporters. This interaction promotes PIN1-polarization to the periclinal membranes of cells in the LRP and establishment of the auxin maximum at its tip. While this work links CPK29 function to fine-tuning PIN-mediated transport of auxin, its induction by nitrate, similarly to other CPKs, remains to be tested [85].

At the level of auxin perception and signaling, the AUXIN SIGNALING F-BOX 3 receptor (AFB3) is a key component of the network integrating auxin and nitrate signaling to control root system adaptations to nitrate availability. *AFB3* expression is induced by nitrate in NRT1.1 transport-dependent manner and repressed by nitrate metabolites through feedback inhibition regulated by *miR393* [11,12]. Downstream of AFB3, the transcription factors NAC4 and OBP4 mediate nitrate-regulated lateral root development [11,12] (Figure 1).

## 4. The Role of Cytokinin in Nitrate-Regulated Plant Growth

Cytokinins belong to a family of N6-substituted adenine derivatives that affect many aspects of plant growth and development including fundamental cellular processes such as cell division and differentiation, as well as embryogenesis, establishment of shoot and root system architecture, apical dominance, phyllotaxis, senescence and others [86,87]. Isopentenyl adenine (iP) and trans-zeatin (tZ) are the two most abundant active cytokinin species in *Arabidopsis* [88,89]. The enzyme isopentenyltransferase (IPT) catalyses the first step of the cytokinin biosynthetic cascade by attaching a prenyl side chain to the N6 position of ADP or ATP. The cytokinin content in *Arabidopsis* is tightly regulated by biosynthetic and degradative processes, as well as by reversible and irreversible conjugation [90,91].

In contrast to the extensively characterized auxin transport system, the building blocks and nature of the cytokinin transport system are still largely unknown [92]. Unlike for auxin, there are no dedicated cytokinin transporters that facilitate cell-to-cell movement. Two families of membrane transporters with cytokinin-transport activity have been identified however, namely, purine permeases (PUP) and equilibrative nucleoside transporters (ENT) [91,93,94]. Recently, PUP14 and its role in the cytokinin transport were thoroughly studied. It was shown that PUP14-mediated cytokinin transport to the cytosol deprives the apoplast of active cytokinin, thus preventing signaling from plasma membrane cytokinin receptors. Importantly, the work demonstrated that the cytokinin response pattern determined by PUP14 activity is essential for plant developmental processes including embryogenesis, lateral root organogenesis, root apical meristem [95,96]. Furthermore, cytokinins transported through the plant in the form of free bases and inactive cytokinin ribosides play an important role as long-distance signaling molecules. In a recent study by Ko et al. [97], ABCG14, a member of ABC transporter family, was identified as a long-distance transporter required for cytokinin translocation from root to shoot. Plants with disruptions in ABCG14 were severely impaired in translocation and distribution of tZ-type cytokinin species synthesized in roots, which subsequently resulted in significant morphological changes in root and shoot growth and development [97,98].

In target cells, cytokinin is perceived by receptors of the histidine kinase (HK) family. In *Arabidopsis*, AHK2, AHK3 and AHK4/CRE1 were recognized for their cytokinin receptor function. Upon cytokinin binding to the receptor its kinase domain is activated and triggers a signal transduction cascade leading to phosphorylation of histidine phosphotransfer proteins (AHPs) and the downstream type-B response regulators (type-B ARRs). The type-B ARRs transcriptional factors bind to a consensus DNA cytokinin motif and trigger expression of cytokinin early response genes. Among them, cytokinin induced type-A ARRs provide a negative feedback to fine-tune cytokinin signaling. The molecular mechanism underlying type-A ARR mediated suppression of cytokinin is scarcely understood. It was proposed that type-A ARRs compete with type B-ARRs for the phosphoryl group while lacking the DNA-binding domain [94].

Nitrate-regulated plant development is tightly linked with cytokinin activity. Nitrate has been shown to promote expression of several genes involved in the cytokinin biosynthesis including *IPT3*, *IPT7* and *CYP735A*2, encoding for cytochrome P450 monooxygenase, required for the biosynthesis of tZ [17,18,19,20,99,100,101,102]. Microarray analysis showed that nitrate-enhanced expression of *IPT3* is mediated in part by NRT1.1 [100]. These results are supported by the finding that cytokinin levels in roots increase within hours after nitrate administration [17,18,19,20].

Besides its effects on cytokinin biosynthesis and levels of biologically active cytokinin derivatives, fine-tuning cytokinin activity through modulating long-distance and cellular cytokinin transport might represent another important regulatory level of nitrate signaling. Cytokinins have been identified as one of the central long-distance signals that mediate nitrogen status relative to nitrogen supply [18,103,104]. Nitrate promotes cytokinin biosynthesis in roots (Figure 2) and their translocation to shoots, where cytokinin acts as positive regulator of shoot growth including leaf expansion and shoot apical meristem activity [21,104]. In roots, cytokinins play a key role in integrating a systemic N signal by adapting the root architecture and fine-tuning nitrate uptake via modulating the expression of key nitrate transporters [20,102]. Recent transcriptomics and ChIP-seq analyses have shown that *PUP14*, and *PUP18*, transporting cytokinin into cells and *ABCG14*, the transporter, responsible for loading cytokinin into the xylem, are among the potential targets of NLP7 [22,27,95,97,98]. While ABCG14 has been found to promote cytokinin translocation from root to shoot on nitrate [18,19,96,97,98] the function of PUP14 and PUP18 in plant growth adaptation to nitrate remains to be elucidated. Additionally, these findings await further confirmation using molecular and genetic tools.

Inevitably, nitrate-modulated cytokinin biosynthesis and transport affect cytokinin signaling. Numerous components of the cytokinin signal transduction pathway, including type-A response regulators, such as *ARR3*, *ARR5* and *ARR6* as well as *Cytokinin Response Factors* (*CRF*s) were recovered as nitrate-responsive genes [10,21,100]. Notably, recent studies pointed at CRFs as an important regulatory hub of convergence of nitrate and cytokinin- and auxin-regulated plant development [10,19,21,22,27,100,105].

## 5. Auxin-Cytokinin Crosstalk in Plant Adaptation to Nitrate Availability

Hormonal pathways interact at multiple levels, thereby forming powerful regulatory networks that maintain homeostasis of the biological systems while integrating inputs from the environment and translating them into adequate adaptive responses. Crosstalk between auxin and cytokinin, two central hormonal regulators of fundamental cellular processes such as cell division and differentiation, governs developmental plasticity—one of the major determinants of the adaptive capacity of the plant body. Thus, further unraveling of the auxin-cytokinin interplay in context of fluctuating environmental conditions such as nitrate availability is essential to gain a better understanding of the mechanisms underlying plant adaptive responses. Recent studies focusing on auxin-cytokinin crosstalk revealed different types of molecular connections. Auxin has been found to promote cytokinin biosynthesis through direct transcriptional control of *IPT* genes mediated by ARF19 [106]. Both auxin and cytokinin signaling pathways are interconnected via *ARF5*-mediated transcriptional control of *CRF2* [107]. Cytokinin fine-tunes auxin distribution via modulation of the polar auxin transport machinery. At transcriptional and posttranslational levels, cytokinin regulates auxin influx and efflux carriers of the Aux/LAX and PIN families, respectively [108,109,110,111,112].

As shown by a recent study, some of those interactions are highly relevant in the context of nitrate regulated plant development as a core component of the molecular framework orchestrating shoot developmental processes with the root nitrate sensory system. It was shown that in response to nitrate supply, NLP7 promotes the expression of cytokinin biosynthesis genes and facilitates cytokinin translocation to shoots. There, CRFs act as direct regulators of *PINs*, stimulating the transport of auxin and thereby promoting shoot growth and development [19].

How the complex auxin-cytokinin regulatory network responds to the availability of nitrogen and coordinates developmental adaptations requires further investigation. One of the few studies that have investigated such complex interactions was conducted by Ristova et al. [113]. In this work, the authors analyzed short-term transcriptional and long-term phenotypic responses to changes in nutrient provision and hormonal treatments. The authors employed a comprehensive set of treatments of nitrogen-starved plants. In addition to nitrate and ammonium supplementation, plants were treated with auxin, cytokinin, and abscisic acid. Setting this work apart from simple treatments, the authors used all possible combinations of these five nutrients/signaling molecules, resulting in a staggering 32 conditions. This allowed Ristova et al. [113] to create a multivariate network that associated nitrogen source and hormone treatment with gene expression and several root traits. Not only did they find many genes already known to be involved in root development, but they were also able to experimentally demonstrate the predictive power of this valuable data resource [113].

Driven by curiosity about nitrogen-source dependent auxin-cytokinin crosstalk we searched Ristova’s dataset for genes with differential sensitivity to auxin, cytokinin or the combined hormonal treatment in dependence of the nitrogen source. Intriguingly, several genes that follow such patterns could be identified (Figure 3). The cytokinin biosynthesis gene *IPT7* shows higher auxin- sensitivity in the presence of ammonium as nitrogen source (Figure 3a). Type-C *ARR22* displays a similar, even more pronounced, expression pattern. These data hint at nitrogen source specific crosstalk between auxin, and cytokinin biosynthesis and signaling. On the other hand, *TAR2* expression (Figure 3b) responds to cytokinin treatment with higher sensitivity when nitrate is available, suggesting a positive effect of cytokinin on auxin biosynthesis in presence of the optimal nitrogen source. Moreover, expression of some genes exhibited nitrogen-source dependent responses to combined treatment with auxin and cytokinin (Figure 3c). Following auxin-cytokinin treatment IAA16 is downregulated when nitrate is present and upregulated in presence of ammonium, further suggesting N-source dependent hormonal crosstalk. Interestingly, *LRL3*, a transcription factor in root hair development, which was previously shown to be upregulated by auxin [114] appears to show this response only in presence of nitrate. Conversely, the expression of *SCARECROW-LIKE 13* (*SCL13)*, a member of the GRAS gene family, responds positively to combined auxin and cytokinin treatment in presence of ammonium and negatively when nitrate is available. Together, these observations provide a hint on as of yet unexplored mechanisms of hormonal crosstalk in the context of nitrogen source availability and strongly warrant further investigation.

The data published in studies like Ristova et al. [113] are a valuable tool and resource for researchers and open up further research opportunities. Combining these studies with the use of experimental platforms such as the split root experiments as in [101] and/or grafting will help to shed more light on the involvement of auxin and cytokinin in the adaptive responses regulated by nitrate in shoot and root. Building on such experiments and data will lead us to a true understanding of the tango that auxin, cytokinin and nitrate are dancing.

## 6. Conclusions and Future Directions

Nitrogen is an important nutrient for plants, so modern agriculture relies heavily on its use as a fertilizer to achieve crop yields of high quantity and quality. However, 67% of nitrogen fertilizers are reportedly not taken up by plants and released into the environment, which has disastrous environmental consequences and hinders cost-effective agricultural practices [115]. For a long time, the research main focus has been on understanding the molecular mechanisms of nitrogen uptake, distribution and assimilation in order to develop strategies to improve nitrogen utilization. Recently, it has become clear that pathways controlling plant nitrogen status closely communicate and interact with hormonal pathways, key coordinators of plant growth and developmental processes. Understanding these interactions and molecular networks between nitrogen and hormones may provide important basis for developing new biotechnologies to minimize the use of nitrogen fertilizers while maximizing the desired outcome hence reducing cost of agriculture and environmental concerns. For example, it is reported that provision of nitrate leads to an increase in biosynthesis and transport of cytokinin to shoots, where it is critical for early shoot adaptations to nitrate. It remains to be seen whether reducing nitrate supply to a critical level while administering cytokinin will result in the same desired growth outcome.

## Figures and Tables

**Figure 1 cells-12-01613-f001:**
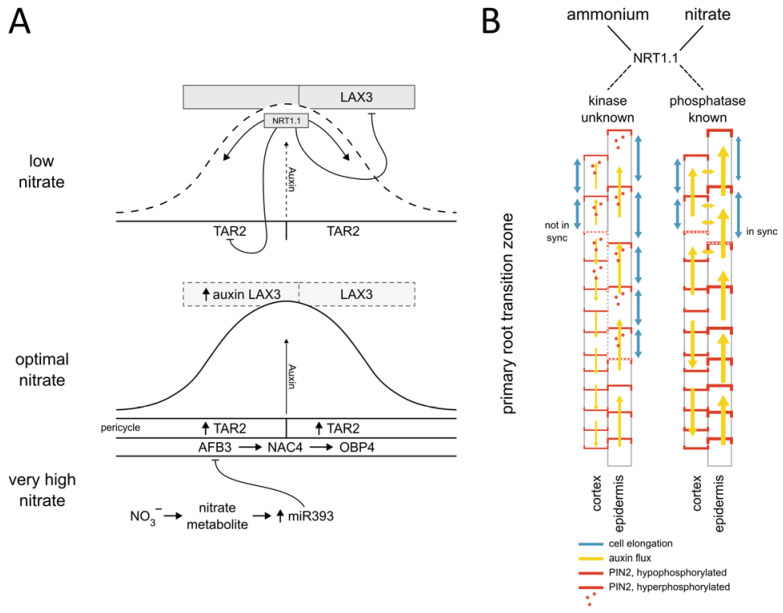
Nitrate modulated auxin synthesis, transport, and signaling converge at regulation of the lateral root primordia (LRP) development and Primary root growth. (**A**) At low nitrate levels NRT1.1 controls the LRP development through transporting auxin and by suppressing TAR2-mediated auxin biosynthesis and LAX3-dependent influx of auxin to cells adjacent to the LRP. The nitrate-dependent expression of the auxin receptor AFB3 is part of the regulatory module controlling LRP development. Under optimal nitrate conditions AFB3 mediates the expression of NAC4 and OBP4 transcription factors, while nitrate metabolite-induced expression of miR393 suppresses AFB3, thus providing a negative feedback loop. (**B**) Nitrogen source-dependent phosphorylation of PIN2 determines the membrane localization of this auxin transporter in epidermal and cortex cells at the primary root (transition zone depicted). The fine-tuning of PIN2 levels at the plasma membrane regulates the auxin flux between two adjacent tissue layers (yellow arrows), thereby coordinating cell elongation patterns (Modified from [9]).

**Figure 2 cells-12-01613-f002:**
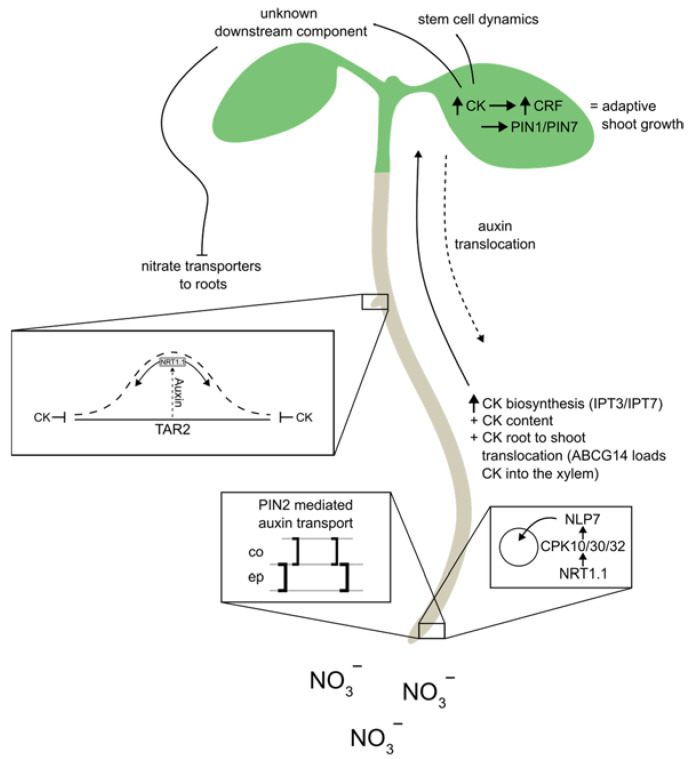
Nitrate regulates the biosynthesis of cytokinins in the root and its translocation from the root to the shoot via NLP7. There it promotes shoot growth through increasing expression of auxin efflux carriers, and induces a systemic downstream unknown component (shoot to root) that regulates nitrate transporter expression in roots, acting as a nitrate satiety signal by inhibiting NRT2.1 expression. In the roots, nitrate regulate Auxin biosynthesis (TAR2), and transport (PIN2, NRT1.1) in addition to cytokinin biosynthesis (IPT3, IPT7) and transport (ABCG14). These numerous auxin and cytokinin pathways that are regulated by nitrate within the roots suggests the possibility of Auxin-Cytokinin crosstalk and its importance in fine-tuning these processes.

**Figure 3 cells-12-01613-f003:**
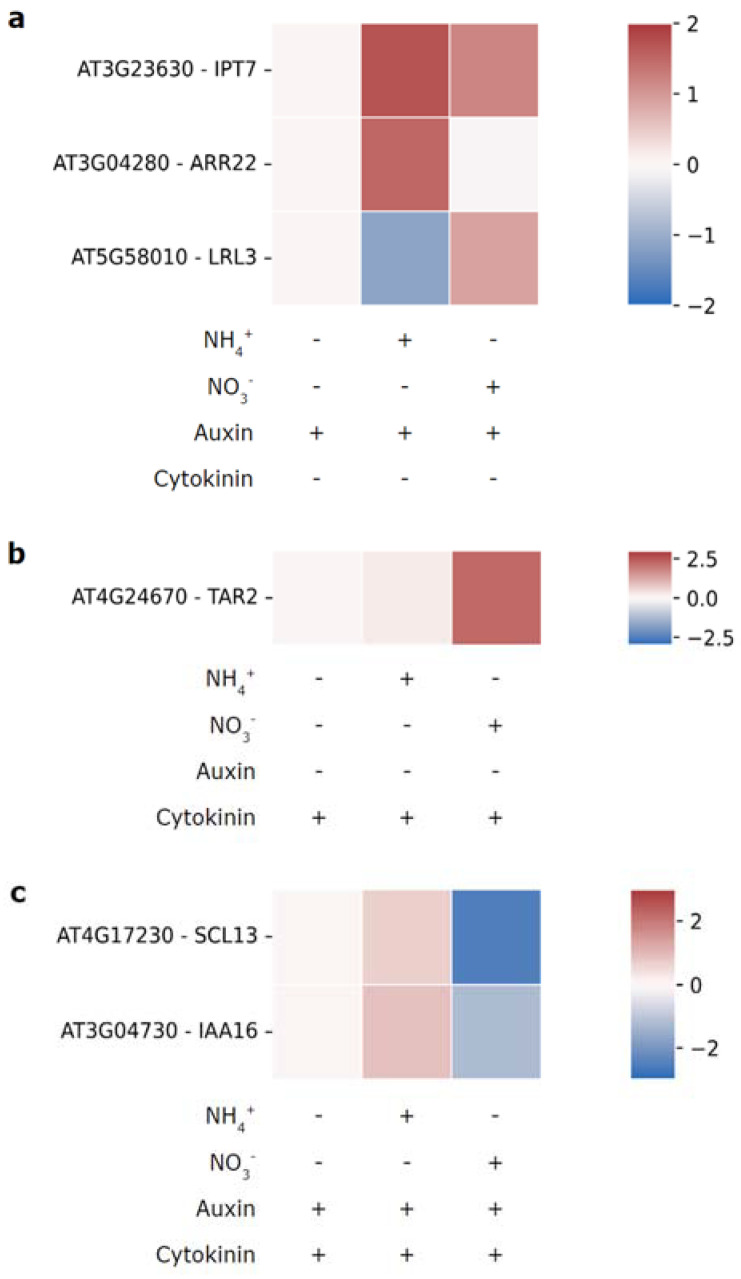
Nitrogen-source-dependent transcriptional effects of auxin (**a**), cytokinin (**b**) and combined hormone treatment (**c**). Following N starvation plants were provided with 1 mM NH_4_^+^, 1 mM NO_3_^-^ or neither, and treated with 0.5 nM of auxin and/or cytokinin. Heatmaps show log2FC in relation to the respective N starved control. Data from [113] (GSE71737).

## Data Availability

Not applicable.

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
