# Peer review of "Nitrate, Auxin and Cytokinin—A Trio to Tango"

_cells, 2023, doi:10.3390/cells12121613_

Round 1
Reviewer 1 Report
Lots of studies and reviews on nitrate/nitrogen and hormonal networks in the last decades. This review focuses on recent advances in the crosstalk in nitrate, auxin, and cytokinin, the new findings and possible missing links. This is an informative and readable review.
It’s a good review, only some writing needs to correct,
1) L257, reference 96 cite twice;
2) Fig 2, stem cell, not stemcell;
3) L382, miss another bracket;
4) Some references need update
Author Response
We would like to thank the Reviewers for their critical review of the manuscript and constructive suggestions.
All minor changes requested by this reviewer have been updated and corrected.
Reviewer 2 Report
Nitrate, auxin, and cytokinin are all critical factors that regulate plant growth and development. Within this context, Rashed Abualia et al. wrote a review focusing on the roles and interactions among these three factors. expound upon the current research delineating nitrate transport and signaling, the involvement of auxin or cytokinin in nitrate-regulated plant growth and development, as well as the auxin-cytokinin crosstalk in plant adaptation to nitrate availability. The authors effectively outlined the significance of nitrogen, auxin, and cytokinin in plant growth and development. They provided a comprehensive and well-structured review of the current understanding of nitrogen-regulated plant growth, with a particular focus on the roles of auxin and cytokinin in plant adaptation to fluctuating nitrogen conditions.
While the authors have successfully covered a wide range of relevant studies, there is room for improvement in highlighting the gaps in our current knowledge and the potential avenues for future research. Additionally, the figures within the review necessitate better organization. Furthermore, the authors might consider discussing the potential applications of this knowledge in agricultural practices, such as improving nitrogen use efficiency in crops and reducing the environmental impact of nitrogen fertilization. This would provide the reader with a broader perspective on the significance and implications of the research in this field.
In conclusion, the authors have delivered a thorough review of the interplay between nitrogen sensing, auxin, cytokinin, and plant adaptation. By addressing the major and minor points, the review could be further strengthened and published in Cells.
Major points:
1. The authors provided a detailed and informative exposition of auxin biosynthesis, transport, and signaling pathways as well as its involvement in plant adaptation to varying levels and sources of nitrogen in plants. The connections between nitrate availability, TAR2-mediated auxin biosynthesis, and the regulation of lateral root outgrowth are well elucidated, as are the roles of various auxin transporters and their nitrate-responsive regulation. It is recommended that Figure 1 be enriched with additional information, such as PIN2, NLP7, and NRT1.1 under optimal and high nitrate conditions. Furthermore, it is crucial to elucidate the interactions and mechanisms by which nitrate affects auxin transporter phosphorylation and how this regulation modulates root growth. The specific roles and functions of various transporters, such as PUP14, PUP18, and ABCG14, in nitrate-regulated cytokinin transport and their contributions to plant development should also be clarified.
2. The function of NRT2.1 (figure 2) has not been adequately addressed in the manuscript. Moreover, it’s not clear how nitrate regulates the biosynthesis of cytokinin in the root and its translocation from the root to the shoot in figure 2. It’s better to add more details in the figure and legend.
3. It’s better to devise a model figure illustrating the auxin-cytokinin crosstalk in plant adaptation to nitrate availability.
4. The authors are suggested to discuss the potential implications of these findings between Nitrate, auxin, and cytokinin for agricultural practices, crop improvement, and environmental sustainability.
5. It is essential to expand upon more discussion on the possible directions for future research.
Minor points:
1. Line 80, 142, 152, 210, 242, 244, 267, should the “Arabidopsis” be italic?
2. Line 81, should the “NPF;” be deleted?
3. Line 84, “73” should be replaced by “72” (53+7+7+5=72).
4. Line 85, “(reviewed by[39,40])” should be replaced by “[39,40]”.
5. It may be necessary to write the full name of a gene or protein for the first time appeared in the text, such as ABCB?
6. Line 276, “(reviewed by[94])” should be replaced by “[94]”.
7. Line 373, “5. Conclusions” should be replaced by “6. Conclusions”.
8. Line 314, the “the a major determinants of the adaptive capacity of the plant body” should be replaced by “the major determinants of the adaptive capacity of the plant body”.
9. Line 341, the “et al” should be replaced by “et al.”.
10. Line 381-384, please rewrite the “The use of elaborate experimental platforms such as the split root as in[100] (or/and grafting and the combination of these two methods will help to shed more light on the involvement of auxin and cytokinin in the adaptive responses regulated by nitrate in shoot and root”. It’s confusing.
11. Please double-check the format of References carefully, such as reference 56.
Author Response
We would like to thank the Reviewers for their critical review of the manuscript and constructive suggestions.
Reviewer 2:
Major points:
- The authors provided a detailed and informative exposition of auxin biosynthesis, transport, and signaling pathways as well as its involvement in plant adaptation to varying levels and sources of nitrogen in plants. The connections between nitrate availability, TAR2-mediated auxin biosynthesis, and the regulation of lateral root outgrowth are well elucidated, as are the roles of various auxin transporters and their nitrate-responsive regulation. It is recommended that Figure 1 be enriched with additional information, such as PIN2, NLP7, and NRT1.1 under optimal and high nitrate conditions. Furthermore, it is crucial to elucidate the interactions and mechanisms by which nitrate affects auxin transporter phosphorylation and how this regulation modulates root growth. The specific roles and functions of various transporters, such as PUP14, PUP18, and ABCG14, in nitrate-regulated cytokinin transport and their contributions to plant development should also be clarified.
Response: We have modified Figure 1 to include PIN2 in the primary root and NRT1.1 control of this process and indicated that kinases/phosphatases downstream of NRT1.1 are still unknown. In addition, the specific roles of ABCG14, PUP14, and PUP18 in the context of cytokinin (CK) were discussed. ABCG14, as a transporter that transports CK into the xylem, thereby supporting CK translocation from the root to the shoot, is highlighted in Figure 2. PUP14 and PUP18 were predicted to be regulated by NLP7. However, the details of their expression modulation by nitrate are still largely unknown, making it difficult to predict their role in the nitrogen-relevant context. Therefore, we have speculated on this in agreement with what we know about their role in CK signal transduction.
2.The function of NRT2.1 (figure 2) has not been adequately addressed in the manuscript. Moreover, it’s not clear how nitrate regulates the biosynthesis of cytokinin in the root and its translocation from the root to the shoot in figure 2. It’s better to add more details in the figure and legend.
Response: CK in the shoot via an unknown molecular mechanism from the shoot to the root inhibits the expression of nitrate transporters in the root and thus acts as a nitrate satiety signal. We also added a paragraph discussing the NRT2.1 function. We agree with the reviewer that this was unclear and has been changed in Figure2.
- It’s better to devise a model figure illustrating the auxin-cytokinin crosstalk in plant adaptation to nitrate availability.
Response: Regarding major point 3: While we agree with the reviewer, unfortunately, there is very little information on auxin-cytokinin cross-talk in the context of nitrogen/nitrate adaptive growth, but we think this is a valid point. Therefore, we have modified Figure 2 to include auxin- and CK-dependent processes regulated by nitrate and their proximity within the root and thus the high likelihood of their interaction.
4.The authors are suggested to discuss the potential implications of these findings between Nitrate, auxin, and cytokinin for agricultural practices, crop improvement, and environmental sustainability.
- It is essential to expand upon more discussion on the possible directions for future research.
Response: Major points 4 and 5: have been included in the conclusions and future directions (section 6). We agree with the reviewer and have speculated how applying molecular studies to develop techniques in agriculture could help improve nitrogen use efficiency.
Minor points:
All points were amended.
Point 11. We re-checked format of all references, concerning the reference 56(old version, 58 current version), as it is a book that contains a review of nitrate transporters, the format slightly differs.
All changes made are highlighted in yellow.
Round 2
Reviewer 2 Report
Thanks for the authors' effort in addressing my questions. I agree that the revised manuscript can be published in Cells.